# Service User and Carer Views and Expectations of Mental Health Nurses: A Systematic Review

**DOI:** 10.3390/ijerph191711001

**Published:** 2022-09-02

**Authors:** Nompilo Moyo, Martin Jones, Diana Kushemererwa, Noushin Arefadib, Adrian Jones, Sandesh Pantha, Richard Gray

**Affiliations:** 1School of Nursing and Midwifery, La Trobe University, Melbourne, VIC 3086, Australia; 2Victorian Tuberculosis Program, Melbourne Health, Melbourne, VIC 3000, Australia; 3Department of Rural Health, University of South Australia, Whyalla Campus, Whyalla Norrie, SA 5608, Australia; 4IIMPACT in Health, University of South Australia, Adelaide, SA 5000, Australia; 5Faculty of Life Sciences, Wrexham Glyndwr University, Wrexham LL11 2AW, UK

**Keywords:** service users, carers, mental health nurses, views, expectations

## Abstract

Service users’ views and expectations of mental health nurses in a UK context were previously reviewed in 2008. The aim of this systematic review is to extend previous research by reviewing international research and work published after the original review. Five databases were searched for studies of any design, published since 2008, that addressed service user and carer views and expectations of mental health nurses. Two reviewers independently completed title and abstract, full-text screening and data extraction. A narrative synthesis was undertaken. We included 49 studies. Most included studies (n = 39, 80%) were qualitative. The importance of the therapeutic relationship and service users being supported in their personal recovery by mental health nurses were core themes identified across included studies. Service users frequently expressed concern about the quality of the therapeutic relationship and indicated that nurses lacked time to spend with them. Carers reported that their concerns were not taken seriously and were often excluded from the care of their relatives. Our critical appraisal identified important sources of bias in included studies. The findings of our review are broadly consistent with previous reviews however the importance of adopting a recovery approach has emerged as a new focus.

## 1. Introduction 

Nurses represent approximately 44% of the global mental health workforce [1]. Mental health services are increasingly focusing on providing patient-centred care and treatment [2,3]; this requires that there is a deep understanding of the views and expectations of people experiencing mental ill-health about mental health nurses [4]. 

Adopting a recovery approach requires nurses to work in a collaborative way with service users to support their recovery objectives [5]. Understanding the views and expectations of service users and their carers may inform mental health policy and practice. In addition, treatment and care for a person suffering from mental ill-health may require a comprehensive and evidence-based approach that encourages service users and their carers to actively participate in their care [6]

The views and expectations of service users towards mental health nurses have been previously reviewed. A systematic review which was undertaken as part of the chief nursing officer’s review of mental health nursing in England (United Kingdom Department of health, 2006) by Bee et al. [7] included 132 studies involving 36,793 participants. The aim of the review was to examine service users’ and carers’ views and expectations of mental health nurses registered and practicing in the United Kingdom. Primary research of any type, where fieldwork was conducted in the UK and was published prior to 2005, was included in the review. The authors undertook a narrative synthesis of included studies and reported that service users viewed mental health nursing as a multifaceted profession that provides practical and social support as well as formal psychological treatments [7]. Review authors also reported consistent negative views of mental health nurses; specifically, that they were inaccessible, did not give enough information to service users, and often did not work in a way perceived as collaborative or patient-centred [7]. The review authors aimed to examine carers’ views and expectations of mental health nurses but could not identify any relevant studies [7].

The methodological quality of included studies in Bee et al. [7] review was determined using guidelines for reviewing non-randomised, observational and qualitative literature [8]. Important methodological limitations were identified across studies that included possible selection bias, and some survey instruments were not assessed for validity [7].

Bee et al. [7] limited the study setting to the United Kingdom and excluded evidence from other countries. To fully understand how people who use mental health services view and perceive MHNs, it would be informative to undertake an updated systematic review of all relevant research, regardless of where fieldwork was conducted. In part, this is because there have been substantial changes to how mental health care is provided since 2005. For example, a shift to recovery-oriented working in, the UK, Australia and North America, from around 2009 [9]. 

We searched Medical Literature Analysis and Retrieval System Online [MEDLINE] [OVID] and Cumulative Index to Nursing and Allied Health Literature [CINAHL] (EbscoHost) from 2005 to 2022 to identify if a review—of any type—that examined the views and expectations of service users and/or carers about mental health nurses had been undertaken. No reviews were identified. 

Aim: To extend and update the systematic review by Bee et al. [7] about the views and expectations of mental health service users and carers about mental health nurses.

## 2. Materials and Methods

The methodology of this systematic review complies with the Preferred Reporting Items for Systematic Reviews and Meta-Analyses 2020 checklist (PRISMA) [10]. As far as possible, we used the methodology described by Bee et al. [7], where we have deviated, we note this in our reporting. The protocol for this review was registered with Open Science Framework (https://doi.org/10.17605/OSF.IO/QWDHU) (accessed on 26 July 2022) after searches were undertaken but prior to data analysis. 

### 2.1. Eligibility Criteria

We included studies that met the following inclusion criteria:Reported the views of service users and carers towards mental health nurses.Were conducted from July 2005 to December 2021. The July 2005 cut-off date was chosen to coincide with the last date of the search for a similar review [7].Were written in English.

No restrictions were placed on the study design, fieldwork settings, or age of participants.

We did not include studies that reported individual case studies, were focused on formal therapeutic interventions (e.g., cognitive behavioural therapy) or evaluated goal-directed nursing tasks (e.g., care planning, medication supervision). In this review, we defined a mental health nurse as a registered nurse working in any mental health setting. The views and expectations of service users/carers were defined as any expressed opinions regarding any aspect of the mental health nurse–service user/carer relationship that happens outside formal therapeutic procedures or task-directed nursing interventions [7].

### 2.2. Information Sources

We searched five electronic databases (platform in brackets):Medical Literature Analysis and Retrieval System Online [MEDLINE] (OVID)Cumulative Index to Nursing and Allied Health Literature [CINAHL] [EbscoHost]Excerpta Medica database [EMBASE] (OVID)Cochrane central (WILEY)PsychINFO (OVID)

We did not search for grey literature—a deviation from the Bee et al. [7] methodology—because the work has not been through a transparent peer-review process [11]. Our initial search was conducted on the 1 June 2020 and updated on the 21 February 2022. 

### 2.3. Search Strategy 

We used the search terms described by Bee et al. [7]. We searched for the Medical Subject Headings (MesH) and free-text phrases such as synonyms or abbreviations. Our review focused on three concepts: (that broadly align with those used by Bee et al. [7]): 1. mental health service users and their family and friends (carers), 2. mental health nurses, 3. views and expectations. Each concept’s MeSH and free-text terms were combined using the Boolean operator ‘OR.’ The Boolean operator ‘AND’ was used to connect all three concepts. The citations were exported from bibliographic databases to Endnote (reference management software). References were then exported to Covidence, a systematic review management software package. The search strategies are shown in Appendix A.

### 2.4. Selection Process 

Title and abstract and full-text screening was undertaken using the Covidence software package by two reviewers (NM, DK, NA, AJ, SP) any discrepancies resolved by a third. 

### 2.5. Data Collection Process 

We extracted data from included studies based on recommendations from the Cochrane handbook [12]. We were not able to use the data items from the Bee et al. [7] review as there were not reported in the manuscript. Two reviewers (NM, DK, NA, AJ, SP) independently extracted data from included studies, a third reviewer resolved any inconsistencies.

### 2.6. Data Items

The following data were extracted from each included study: author (coded surname, initial), year of publication, digital object identifier, the country where fieldwork was conducted, study aim, study setting (coded inpatient, community, inpatient and community (mixed), study population(s) (coded service users, carers), study design, sampling method, data collection procedures (coded interviews, focus groups, surveys), psychometric properties (validity) of measures used, approach to data analysis, summary of key study findings.

### 2.7. Risk of Bias Assessment 

Quality appraisal of included studies was undertaken using three measures (Effective Public Health Practice Project Quality Assessment Tool (EPHPP), Mixed Methods Appraisal Tool (MMAT), and Joanna Briggs Institute (JBI) critical appraisal checklist for qualitative research]. The use of formal quality appraisal measures is divergent from the Bee et al. [7] review that did not use recognised critical appraisal tools. 

We used the EPHPP to assess the risk of bias for observational studies [13]. The EPHPP has established psychometric properties [13,14]. The tool has six components: 1. selection bias, 2. study design, 3. confounders, 4. blinding, 5. data collection method, 6. withdrawals and dropouts [13], each appraised as strong, moderate, or weak. The overall rating of the study is determined based on the following criteria: strong (no weak ratings), moderate (one weak rating) or weak (two or more weak ratings) [13]. 

For mixed method studies, we used the MMAT [15]. The measure contains five criteria: 1. the rationale for using mixed methods, 2. integration of qualitative and quantitative study components, 3. interpretation of the results, 4. divergences and inconsistencies between quantitative and qualitative findings, and 5. different study components adhering to the quality criteria of each methodological tradition. Each item is rated “yes”, “no”, “cannot tell”, [15]. The MMAT does not produce an overall quality rating. 

JBI critical appraisal tool for qualitative research [16] has ten items rated “yes”, “no”, “unclear”, or “not applicable”. The items are: 1. research methodology and its philosophical perspective, 2. study design, 3. data collection method, 4. data analysis method, 5. interpretation of results, 6. cultural or theoretical location of the researcher, 7. the influence of the researcher on the study, 8. representation of participants and their perspectives, 9. ethical considerations, and 10. the relationship between the results and the participants’ views [16]. Items are reported as a summary, there is no overall quality rating. 

Two reviewers completed the critical appraisal task independently. A third reviewer resolved any disagreements between reviewers. 

### 2.8. Grouping Studies for Synthesis

We made a post hoc decision to group included studies based on study population: 1. service users, 2. carers. 

In our protocol, we also stated that we would group studies based on the clinical setting where fieldwork was undertaken, this approach did not prove practical because of the large number of clinical settings we identified. We, therefore, made a post hoc decision to recode studies into three broader clinical groupings: 1. hospital inpatient, 2. community, 3. mixed [community and inpatient] services.

### 2.9. Data Synthesis

We summarized the findings of multiple primary studies using words and text, a technique known as narrative synthesis [17]. We extracted data from the included studies in tabular form and critically appraised the methodological quality of each study [see Table 1, Table 2, Table 3 and Table 4]. We briefly described each study to familiarize ourselves with the research and compare findings. Studies were grouped based on the views of service users and carers about MHNs, this made it easier to describe, analyze, and look for patterns within and across these groups. Emerging themes across the studies were identified. In 38 studies, in-depth interviews were conducted to collect data, yielding varying themes. These themes occasionally overlapped, necessitating the classification of some studies under multiple themes. Content and thematic analysis were mixed to comprehensively describe the findings from included studies. We could not perform a meta-analysis of the survey and experimental studies because of the heterogeneity of methodologies and outcomes. 

### 2.10. Amendments to Information Provided at Registration

We made three amendments to the study following the registration of the protocol with the Open Science Framework. Amendment one was a change to the study title, which was originally described as an updated systematic review and which we changed to a systematic review. We also amended the review aim and method to reflect this amendment. The Mixed Methods Appraisal Tool was listed in the protocol as the quality appraisal measure for all included studies. We considered that other quality appraisal measures were more appropriate for determining the risk of bias in included studies. Post hoc, we decided to use the EPHPP for observational and experimental and the JBI appraisal checklist for qualitative studies. The Mixed Methods Appraisal Tool was retained for mixed methods only. As stated above, during study synthesis, we coded studies into three clinical groupings. 

## 3. Results

### 3.1. Description of Included Studies

The flow of studies through the review is shown in Figure 1. Our search generated 26,938 studies. Fifty papers, reporting 49 studies, met our inclusion criteria and were included in the review. Authors of one study reported results across two papers [30,31]. Appendix A is a complete list of all included articles. Studies excluded at full-text screening (n = 27) [68,69,70,71,72,73,74,75,76,77,78,79,80,81,82,83,84,85,86,87,88,89,90,91,92,93,94] are listed in Appendix A. Data extracted from included studies are summarised in Table 1. 

We contacted the corresponding authors of nine studies to request additional information on: 1. the clinical setting where fieldwork was conducted [29,30,31], 2. the population and sample size [30,31,34,37,41,55], 3. ethics approval [64] and 4. approach to data analysis [28,40]. As of the 27 June 2022, two authors, [28,37] responded, providing the requested information that we incorporated in the risk assessment data extraction tables, respectively. 

Table 2 is a summary of the characteristics of included studies. Around eight out of ten included studies were described as qualitative and used interviews or focus groups as a method of data collection. Fieldwork for two-thirds of the studies was conducted with participants that were community dwelling. Three-fifths of the studies were conducted in countries with advanced economies, including Australia, the United Kingdom and the United States of America. 

Two-thirds of studies were exclusively focused on service users. Ten studies included both service users and carers, and five just carers. The total number of service users and carers involved in included studies was 1689 and 166 (the number of carers included was not reported in three studies [30,31,34,55] respectively). The median sample size for included studies was 15 (range, five to 511). The diagnosis of participants was reported in half of the studies. The most common reported psychiatric diagnoses were schizophrenia, personality disorders, and depression. 

### 3.2. Quality Appraisal

Table 3, Table 4 and Table 5 summarise the quality appraisal for included studies against the criteria from the relevant critical appraisal measure. All eight observational and experimental studies were rated as having a high risk of bias (Table 3). Six out of eight studies had a risk of selection bias because the authors used convenience sampling to identify participants. Most author used measures were developed specifically for the study [n = 5] that had not been validated. 

The two mixed methods studies satisfied three of the five items of the MMAT (Table 4). None of the included studies addressed divergences or inconsistencies between quantitative and qualitative findings. Giménez-Díez et al. [28] did not provide a compelling rationale for utilising a mixed methods design to address the research question. In the McCloughen et al. [48] study, the authors did not demonstrate the validity of the measure used in the study. 

Table 5 summarises qualitative studies against the ten JBI critical appraisal criteria. Included qualitative studies addressed the majority of JBI criteria. Two criteria that were not addressed by over half of the included studies were statements that contextualised the researcher culturally or theoretically (item 6) and addressed the effect of the researcher on the study and vice versa (item 7). We note that the authors of the two studies did not explicitly state that the study had been reviewed and approved by an ethics committee or Institutional Review Board [23,95].

Brimblecombe et al. [21] conducted a national consultation study using electronic response forms and open meetings to collect data and analysed it using content analysis. We could not identify a relevant critical appraisal tool for this type of research.

Our results are organized under service user and carer views and expectations of mental health nurses.

**Table 5 ijerph-19-11001-t005:** Risk of bias assessment for qualitative studies using the Joanna Briggs Institute checklist for qualitative research.

Study Author	Digital Object Identifier (DOI)	Criteria 1	Criteria 2	Criteria 3	Criteria 4	Criteria 5	Criteria 6	Criteria 7	Criteria 8	Criteria 9	Criteria 10
Ådnøy Eriksen et al. (2014) [18]	https://doi.org/10.1111/inm.12024 (accessed on 20 June 2022)	Yes	Yes	Yes	Yes	Yes	No	No	Yes	Yes	Yes
Askey et al. (2009) [19]	https://doi.org/10.1111/j.1467-6427.2009.00470.x (accessed on 20 June 2022)	Yes	Yes	Yes	Yes	Yes	No	No	Yes	Yes	Yes
Biringer et al. (2021) [20]	https://doi.org/10.1111/ppc.12633 (accessed on 20 June 2022)	Yes	Yes	Yes	Yes	Yes	Unclear	Unclear	Yes	Yes	Yes
Coatsworth-Puspoky et al. (2006) [22]	https://doi.org/10.1111/j.1365-2850.2006.00968.x (accessed on 20 June 2022)	Yes	Yes	Yes	Yes	Yes	No	No	Yes	Yes	Yes
Cunningham & Slevin (2005) [23]	https://doi.org/10.1111/j.1365-2850.2004.00769.x (accessed on 20 June 2022)	Yes	Unclear	Yes	Yes	Yes	No	No	Yes	No	Yes
Earle et al. (2011) [24]	https://doi.org/10.1111/j.1365-2850.2010.01672.x (accessed on 20 June 2022)	Yes	Yes	Yes	Yes	Yes	No	Yes	Yes	Yes	Yes
Evans et al. (2021) [25]	https://doi.org/10.1111/inm.12795 (accessed on 20 June 2022)	Yes	Yes	Yes	Yes	Yes	Yes	Yes	Yes	Yes	Yes
Frain et al.l. (2021) [26]	https://doi.org/10.1080/01612840.2020.1820120 (accessed on 20 June 2022)	Yes	Yes	Yes	Yes	Yes	No	Yes	Yes	Yes	Yes
Gerace et al. (2018) [27]	https://doi.org/10.1111/inm.12298 (accessed on 27 July 2022)	Yes	Yes	Yes	Yes	Yes	No	No	Yes	Yes	Yes
Goodwin & Happell (2006) [29]	https://doi.org/10.1111/j.1447-0349.2006.00413.x (accessed on 27 July 2022)	Yes	Yes	Yes	Yes	Yes	No	No	Yes	Yes	Yes
Goodwin & Happell (2007) [30]	https://doi.org/10.1080/01612840701354596 (accessed on 27 July 2022)	Yes	Yes	Yes	Yes	Yes	No	No	Yes	Yes	Yes
Goodwin & Happell (2007) [31]	https://doi.org/10.1080/01612840701354612 (accessed on 27 July 2022)	Yes	Yes	Yes	Yes	Yes	No	No	Yes	Yes	Yes
Gray & Brown (2017) [32]	https://doi.org/10.1111/inm.12296 (accessed on 27 July 2022)	Yes	Yes	Yes	Yes	Yes	Yes	Yes	Yes	Yes	Yes
Gunasekara et al. (2014) [33]	https://doi.org/10.1111/inm.12027 (accessed on 27 July 2022)	Yes	Yes	Yes	Yes	Yes	No	No	Yes	Yes	Yes
Happell & Palmer (2010) [35]	https://doi.org/10.3109/01612840.2010.488784 (accessed on 27 July 2022) (accessed on 27 July 2022)	Yes	Yes	Yes	Yes	Yes	No	Yes	Yes	Yes	Yes
Horgan et al. (2021) [36]	https://doi.org/10.1111/inm.12768 (accessed on 27 July 2022)	Yes	Yes	Yes	Yes	Yes	No	No	Yes	Yes	Yes
Jones et al. (2007) [37]	https://doi.org/10.1111/j.1365-2648.2007.04332.x (accessed on 27 July 2022)	Yes	Yes	Yes	Yes	Yes	No	Yes	Yes	Yes	Yes
Keogh et al. (2020) [38]	https://doi.org/10.1080/01612840.2020.1731889 (accessed on 27 July 2022)	Yes	Yes	Yes	Yes	Yes	No	Yes	Yes	Yes	Yes
Kertchok (2014) [39]	https://doi.org/10.3109/01612840.2014.908439 (accessed on 27 July 2022)	Yes	Yes	Yes	Yes	Yes	No	Yes	Yes	Yes	Yes
Lees et al. (2014) [42]	https://doi.org/10.1111/inm.12061 (accessed on 27 July 2022)	Yes	Yes	Yes	Yes	Yes	No	No	Yes	Yes	Yes
Lessard- Deschênes, & Goulet (2022) [43]	https://doi.org/10.1111/jpm.12800 (accessed on 27 July 2022)	Yes	Yes	Yes	Yes	Yes	No	No	Yes	Yes	Yes
Lim et al. (2019) [44]	http://hdl.handle.net/20.500.11937/77779 (accessed on 27 July 2022)	Yes	Yes	Yes	Yes	Yes	No	No	Yes	Yes	Yes
McAllister et al. (2021) [45]	https://doi.org/10.1111/inm.12835 (accessed on 27 July 2022)	Yes	Yes	Yes	Yes	Yes	No	Yes	Yes	Yes	Yes
McCann et al. (2012) [47]	https://doi.org/10.1111/j.1365-2702.2011.03836.x (accessed on 27 July 2022)	Yes	Yes	Yes	Yes	Yes	No	No	Yes	Yes	Yes
Moll et al. (2018) [49]	https://doi.org/10.3928/02793695-20180305-04 (accessed on 27 July 2022)	Yes	Yes	Yes	Yes	Yes	No	No	Yes	Yes	Yes
Montreuil et al. (2015) [50]	https://doi.org/10.3109/01612840.2015.1075235 (accessed on 27 July 2022)	Yes	Yes	Yes	Yes	Yes	Yes	Unclear	Yes	Yes	Yes
Pitkänen et al. (2008) [51]	https://doi.org/10.1016/j.ijnurstu.2008.03.003 (accessed on 27 July 2022)	Yes	Yes	Yes	Yes	Yes	Yes	No	Yes	Yes	Yes
Romeu-Labayen et al. (2022) [53]	https://doi.org/10.1111/jpm.12766 (accessed on 14 July 2022)	Yes	Yes	Yes	Yes	Yes	No	Yes	Yes	Yes	Yes
Rose et al. (2015) [54]	https://doi.org/10.1017/S2045796013000693 (accessed on 27 July 2022)	Yes	Yes	Yes	Yes	Yes	No	No	Yes	Yes	Yes
Rydon (2005) [55]	https://doi.org/10.1111/j.1440-0979.2005.00363.x (accessed on 27 July 2022)	Yes	Yes	Yes	Yes	Yes	No	No	Yes	Yes	Yes
Santangelo et al. (2018) [56]	https://doi.org/10.1111/inm.12317 (accessed on 27 July 2022)	Yes	Yes	Yes	Yes	Yes	No	No	Yes	Yes	Yes
Schneidtinger et al. (2019) [58]	https://doi.org/10.1111/jcap.12245 (accessed on 27 July 2022)	Yes	Yes	Yes	Yes	Yes	No	No	Yes	Yes	Yes
Shattell et al. (2007) [59]	https://doi.org/10.1111/j.1447-0349.2007.00477.x (accessed on 27 July 2022)	Yes	Yes	Yes	Yes	Yes	No	No	Yes	Yes	Yes
Stenhouse (2011) [61]	https://doi.org/10.1111/j.1365-2850.2010.01645.x (accessed on 27 July 2022)	Yes	Yes	Yes	Yes	Yes	No	Yes	Yes	Yes	Yes
Stewart et al. (2015) [62]	https://doi.org/10.1111/inm.12107 (accessed on 27 July 2022)	Yes	Yes	Yes	Yes	Yes	No	Yes	Yes	Yes	Yes
Terry (2020) [63]	https://doi.org/10.1111/inm.12676 (accessed on 27 July 2022)	Yes	Yes	Yes	Yes	Yes	No	No	Yes	Yes	Yes
Testerink et al. (2019) [64]	https://doi.org/10.1111/ppc.12275 (accessed on 27 July 2022)	Yes	Yes	Yes	Yes	Yes	No	Yes	Yes	Unclear	Yes
Wilson (2010) [66]	https://doi.org/10.1111/j.1365-2850.2010.01586.x (accessed on 27 July 2022)	Yes	Yes	Yes	Yes	Yes	No	Yes	Yes	No	Yes
Wortans et al. (2006) [67]	https://doi.org/10.1111/j.1365-2850.2006.00916.x (accessed on 27 July 2022)	Yes	Yes	Yes	Yes	Yes	No	Yes	Yes	Yes	Yes

### 3.3. Service Users’ Views and Expectations of Mental Health Nurses 

#### 3.3.1. Satisfaction with Nursing Care 

Nine studies (involving 1009 participants) examined service user satisfaction with mental health nursing care [28,34,35,38,40,41,57,60,65]. Service users generally reported they were satisfied with the mental health nursing care they received. For example, Saur et al. [57] surveyed 105 people with major depression who attended a primary care clinic. The authors stated that 78 [74%] service users assessed the quality of care received as excellent, and 91 [88%] reported being very satisfied with the care received [57].

#### 3.3.2. Service User-Centred Care

Across six studies [33,36,44,48,51,62], service users consistently reported an expectation that MHNs should work in a service user-centred way. For example, Gunasekara et al. [33] interviewed 20 people with lived experience of inpatient psychiatric treatment to explore their perspectives on MHN care. Service users emphasised the importance of planning care around their specific needs and showing an interest in them as individuals [33].

#### 3.3.3. Recovery-Focused Care 

One hundred and forty-three service users participated in eight studies reporting views on how nurses supported their recovery [27,44,50,51,52,55,58,95]. Although participants across these studies did not spontaneously express, that mental health nurses were working in a recovery way, they did report expectations of aspects of care that are consistent with recovery-orientated practices. For example, a qualitative study involving nine participants by Schneidtinger & Haslinger-Baumann, [58] reported that service users viewed MHNs as supportive, accessible, and helped them develop new coping strategies, which they found supportive for their recovery. Another study by Rydon, [55] of 21 people with a mental ill-health diagnosis reported the importance of mental health nurses conveying hope to them as individuals and to their carers. Helping people to look beyond mental ill-health—engaging in hobbies and meaningful occupations—and supporting them in making their own decisions which are again recovery-focused expectations, was reported in the Pitkänen et al. [51]. Thirty-one people with mental ill-health participated in a qualitative study about their perspectives on recovery-focused care and aggression in the inpatient services [44]. Service users reported that engagement by mental health nurses in therapeutic interactions during crisis encouraged self-management of behaviour [a key component of recovery-focused care] [44].

#### 3.3.4. Mental Health Nurse Flexibility

Across four included studies—all with a community focus—there was a narrative that service users viewed MHNs flexibility around the timings of meetings as particularly important and valuable [28,35,58,65]. 

#### 3.3.5. Therapeutic Relationships

Involving 64 community dwelling people, the authors of four studies identified positive views from service users about the quality of their therapeutic relationships with MHNs [18,22,27,53,59]. Gerace et al. [27] and Shattell et al. [59] reported—from two qualitative studies involving seven and 20 participants—that service users reported that they were able to engage with MHNs in meaningful relationships. 

#### 3.3.6. Expectation of Interventions Delivered by Mental Health Nurses

The authors of 13 studies reported on service users’ expectations of the types of interventions delivered by mental health nurses [19,21,25,32,33,50,51,52,54,56,62,63,95]. Providing psychosocial support and fostering hope were identified as core interventions across multiple studies [50,51,52]. In one study, medication administration and assisting with self-care were identified by service users as examples of interventions they expect MHNs to deliver [95].

Gray & Brown, [32] conducted a qualitative study of 15 inpatients about MHNs’ physical health care. Generally, participants reported that physical health care was an important part of the work of MHNs but indicated that they often failed to pay adequate attention to addressing these needs [32]. 

Views of service users about the competencies of MHNs in delivering culturally congruent care were identified in one qualitative study of community dwelling participants [95]. The 15 African American adults viewed MHNs as reassuring, understanding, and supportive of their spiritual needs [95]. 

#### 3.3.7. Important Qualities of Mental Health Nurses

The authors of two studies [36,55] involving 71 people with mental ill health in the community explored the values MHNs. Service users in both studies expected MHNs to treat them with respect, have a good understanding of mental ill-health, be supportive, non-discriminatory, non-stigmatizing, non-judgemental, convey hope, and a willingness to spend time talking with them [36,55]. 

#### 3.3.8. Negative Views of Mental Health Nurses

Across 13 studies, services users reported notable negative views of mental health nurses, which included skills deficits, negative attitudes, and poor therapeutic engagement [20,23,25,32,42,43,44,45,48,55,59,61,62]. For example, in a study involving 119 inpatient service users from a single Mental Health service in England, the authors reported that MHNs seemingly lacked the necessary skills [but did not provide examples] to address their needs [62]. The authors reported that some service users viewed MHNs as uncaring, dismissive, and disrespectful [62]. In another study, Rose et al. [54] interviewed 37 inpatient service users with schizophrenia who reported that they rarely experienced a therapeutic relationship with MHNs on the ward [54]. In another study, inpatient service users viewed MHNs as not supportive and ineffective communicators who were quick to judge them as potentially aggressive when they expressed dissatisfaction with the care they received [44]. Not working in a collaborative way was reported by the authors of two qualitative studies [20,48]. McCloughen et al. [48] used focus groups and surveys of 18 inpatients service users. MHNs were viewed as being inaccessible and inflexible in by study participants [48].

#### 3.3.9. Views of Mental Health Nurse Prescribers

Five studies involving 118 service users focused on service users’ views about the MHN prescribing [24,26,37,46,67]. In all five studies, service users talked about being given a choice and being more involved in decisions about their medication. Service users reported that MHN prescribers took time to build a positive therapeutic alliance and ensure that they provided detailed information about treatment options available to them [24,26,46,67]. Service users described MHN prescribers as supportive and non-judgemental [26], confident [67], prompt, courteous, responsive and thorough in their work, as well as able to communicate effectively [67].

### 3.4. Views and Expectations of Carers

The authors of 15 studies reported views and expectations of carers about MHNs that we have addressed under two headings: collaborating with carers and negative views [19,21,28,29,30,31,33,34,39,41,45,47,49,50,55,64].

#### 3.4.1. Collaborating with Carers 

A consistent expectation from carers that MHNs work collaboratively with them in supporting their relative was identified across seven studies [19,29,30,31,33,39,45,47]. For example, 17 carers were interviewed in a study by Kertchok [39], reporting that MHNs were generally collaborative, understood their concerns, gave them time to talk and provided information about how to care for their relatives [39]. 

#### 3.4.2. Negative Views 

The authors of five studies reported negative views of mental health nurses [29,30,31,45,47,64]. For example, in one study, carers reported frustration with the lack of interactions with MHNs regarding the treatment that their relatives were receiving and expressed concern that their views about how to care for their relatives were not considered [45]. McCann et al. [47] reported that carers considered their role as undervalued by MHNs who excluded them from meetings about care and treatment planning because of concerns about privacy and confidentiality. Carers viewed MHNs as not providing adequate information about service users’ illness and treatment [29,30,31].

Carers in one Australian study (number of participants not provided) reported that they were concerned that new nursing graduates—who had completed comprehensive nurse education—were not adequately prepared to work in psychiatric clinical settings [31].

## 4. Discussion 

The aim of this systematic review was to extend and update the work by Bee et al. [7] that examined the views and expectations of mental health service users and carers about mental health nurses in the UK. We included 49 new studies published after the Bee et al. [7] systematic review. The methodological quality of included studies was generally poor, and the preponderance of qualitative studies limits the generalisability of the observations beyond the context of the research. Our results were consistent with those reported in the Bee et al. [7] review: service users were generally satisfied with the care they received from mental health nurses and considered that they provided good levels of psychosocial support. Across included studies, service users expressed concern about the quality of the therapeutic relationship with mental health nurses and low levels of collaborative working.

These observations are striking, given the profound shift over the past 15 years toward a recovery-orientated and, more recently, trauma-informed way of working [66,96]. Our findings may indicate that the necessary changes in mental health nursing practice have not occurred. Put bluntly, what mental health services say they provide and how nurses practice are starkly different. Our findings may be explained by a failure to support MHNs in developing the necessary competencies or a disregard by MHNs for this way of working.

Our review identified emergent expectations of mental health nurses around promoting recovery and addressing service users’ physical health that were not identified in the previous review. 

Our findings are broadly consistent with other related reviews [97,98]. For example, Newman et al. [97] conducted an integrative literature of service users’ experiences of inpatient and community mental health services that included 34—predominantly qualitative—studies. Authors indicated that service users were rarely involved in decisions about their care [97]. In addition, Newman et al. [97] report that despite the lack of therapeutic relationships between service users and healthcare professionals, service users were satisfied with the quality of care they received, which is consistent with our findings. Eassom et al. [98] identified privacy concerns and confidentiality as barriers to carer engagement in a review of 43 studies involving 321 service users and 276 carers, which is again, consistent with our results. 

The views and expectations of service users and carers about MHNs were similar. In some studies, for example, [31,62], service users and carers reported strikingly negative views about MHNs, which included deficits in fundamental skills to work in clinical environments. Although these were qualitative studies, and consequently, observations cannot be generalised, the research raises concerns about the quality of mental health nursing care service users receive. These findings are dissonant with the high levels of satisfaction reported in observational studies, which may be explained by high levels of social desirability bias in surveys of this type. To generate a deep understanding of the lived experience of mental health nursing care, large multi-centre qualitative research is required.

### 4.1. Limitations of the Evidence Included in the Review 

Included studies had important methodological limitations that need to be considered when interpreting the findings of this review. Authors of surveys included in the review consistently relied on convenience sampling methods, for example, recruiting participants from a single clinical area or inpatient ward. Likely this may introduce selection bias limiting the generalizability of the observations. The authors of many qualitative studies did not locate themselves culturally or theoretically, nor consider how their views and experiences may influence the study findings. 

### 4.2. Limitations of the Review Processes 

There were several limitations to the review process that need to be considered. Firstly, we did not include grey literature or studies not in the English language, which may mean that important work may have been omitted. The protocol for this review was registered with the Open Science Framework (OSF) [99] after initial searches were undertaken, consequently, we cannot demonstrate that our search strategy was not amended after the review had started, and this could be an important source of bias. 

### 4.3. Implications of the Results for Practice 

The implications for practice are limited by the poor methodological quality of included studies. Understanding mental health service users’ views and expectations of nurses requires research that is methodologically rigorous, appropriately powered and uses well validated measures. The preponderance of small qualitative studies adds little by way of contribution to knowledge and may confuse or distort the evidence base. 

## 5. Conclusions

Service users’ and carers’ views and expectations of mental health nurses have not changed much qualitatively over the last 15 years. The emerging theme is that service users expect mental health nurses to provide recovery-focused care and attend to their physical health needs. However, the poor methodological quality of included studies is concerning and means that essentially meaningful conclusions cannot be made. 

## Figures and Tables

**Figure 1 ijerph-19-11001-f001:**
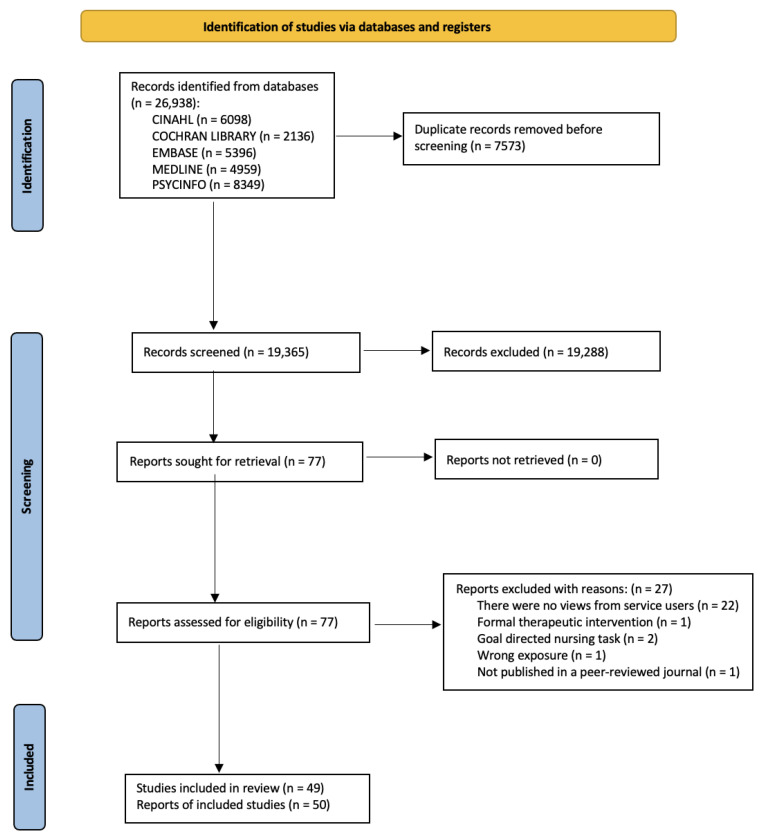
PRISMA flow diagram.

**Table 1 ijerph-19-11001-t001:** Summary of the included studies.

Study Author (Year)	Digital Object Identifier (DOI)	Country Where Fieldwork Was Conducted	Study Aim	Study Setting Where Field Was Conducted	Population(s) under Investigation	Study Design	Sampling Procedure	Data Collection Procedures	Reported Psychometric Properties of Measures (For Cross Sectional, Quasi-Experimental, and Randomised Control Trial Study)	Data Analysis	Summary of Key Study Findings
Ådnøy Eriksen et al. (2014) [18]	https://doi.org/10.1111/inm.12024 (accessed on 10 June 2022)	Norway	To explore how service users perceive their relationships with mental health nurses (MHNs) and how these relationships may hinder or promote recovery	Community Mental Health setting	Service users between 20 and 60 years with serious mental illness (n = 11)	Interpretative phonological analysis	Convenience sampling	One-on-one interviews		Interpretative phenomenological analysis	Service users’ relationships with MHNs were conditional. The individual’s autonomy was reduced when expected to match the MHNs’ expectations. Service users felt safe talking to MHNs who valued their ideas, beliefs, and ambitions.
Askey et al. (2009) [19]	https://doi.org/10.1111/j.1467-6427.2009.00470.x (accessed on 10 June 2022)	United Kingdom	To examine the perspectives and experiences of carers and service users regarding what carers of people with psychosis require from mental health services	Community Mental Health setting (n = 2 centres)	Service users aged between 16 and 64 years (n = 12) Carers (n = 22)	Qualitative study	Convenience sampling	One-on-one interviews and focus group		Thematic analysis	Increasing carer engagement was perceived as critical by all groups. Carers perceive that MHNs should be more respectful and listen to carers. The service users thought carers must be educated about psychosis.
Biringer et al. (2021) [20]	https://doi.org/10.1111/ppc.12633 (accessed on 10 June 2022)	Norway	To examine how service providers collaborate and coordinate to help service users recover	Community mental health setting	Service users with complex and severe mental illness (n = 6)	Qualitative study	Convenience sampling	Group interviews		Thematic analysis	Participants want to be involved in making decisions about the nursing care. Service users expected nurses to be accessible and flexible in their help. They desired MHNs to visit their homes.
Brimblecombe et al. (2007) [21]	https://doi.org/10.1111/j.1365-2850.2007.01119.x (accessed on 10 June 2022)	United Kingdom	To explore the perspectives of multiple stakeholders on how MHNs can improve service users’ experiences and outcomes in inpatient care settings.	Community setting	Service users (n = 11) Carers (n = 3)	Consultation	Convenience sampling	Survey and open meetings		Content analysis	Service users desire more privacy and security. Carers expect MHNs to provide meaningful activities to service users. Service users believe inpatient MHNs need more training.
Coatsworth-Puspoky et al. (2006) [22]	https://doi.org/10.1111/j.1365-2850.2006.00968.x (accessed on 10 June 2022)	Canada	To explore the cultural and contextual factors that influence the development of the nurse-service user relationship.	Consumer-survivor organisations (n = 2 centres)	Service users with mood disorders, panic disorder, personality disorder & schizophrenia (n = 14)	Mini-ethnography design	Convenience sampling	One-on-one interviews and field notes		Thematic analysis	There are two types of relationship. (1) The service user feels accepted by the MHN, is treated with respect, and discloses personal problems. (2) The MHN is not very helpful, and the service user feels frustrated. So, the nurse and the service user ignore one another.
Cunningham & Slevin (2005) [23]	https://doi.org/10.1111/j.1365-2850.2004.00769.x (accessed on 14 July 2022)	Ireland	The aim of the study was not explicitly stated. The author states in the discussion that the study collected the views of service users on the role of the community MHNs	Community mental health setting (n = 2 centres)	Service users with depression, anxiety, eating disorders and schizophrenia (n = 13)	Qualitative	Convenience sampling	Focus group		Thematic content analysis	Some service users considered MHNs helpful. The roles of MHNs and other professionals in multidisciplinary teams were not well understood by service users. Service users perceive that people with lived experiences of mental illness should be included in the multidisciplinary team
Earle et al. (2011) [24]	https://doi.org/10.1111/j.1365-2850.2010.01672.x (accessed on 14 July 2022)	United Kingdom	To find out what service users thought about the care they got from MHN prescribers	Early intervention service	Service users aged between 16 and 35 years with first episode of psychosis (n = 6)	Qualitative case study	Convenience sampling	One-on-one interview		Interpretative phenomenological analysis	Service users preferred getting their medications from MHN prescribers because it was more convenient and less stressful. MHN prescribers gave service users the option of selecting their own medication. The potential benefits of medication were not adequately explained.
Evans et al. (2021) [25]	https://doi.org/10.1111/inm.12795 (accessed on 10 June 2022)	Australia	To explore how service users residing in longer-stay mental health rehabilitation services were able (or not) to negotiate and sustain sexual expression	Mental health rehabilitation setting	Service users aged between 16 and 64 years with schizophrenia and schizo-affective disorder (n = 11)	Qualitative case study	Purposive sampling	One-on-one interviews		Thematic analysis	MHNs policed the facility’s physical area and restricted sexual expression. MHNs entered service users’ bedrooms without warning or permission, violating their right to privacy. Service users felt that sexual expression limits hindered recovery.
Frain et al. (2021) [26]	https://doi.org/10.1080/01612840.2020.1820120 (accessed on 10 June 2022)	Ireland	To examine the experiences of service users with MHN prescribers in a homecare setting	Community mental health setting (n = 2 centres)	Service users aged from 16 to 64 years (n = 12)	Qualitative exploratory	Purposive sampling	One-on-one interviews		Thematic analysis	The MHN prescriber helped the service users to feel heard and understood. Participants highly valued the continuity of treatment provided by the MHN prescriber. The nurse prescription service increased compliance and decreased non-disclosure.
Gerace et al. (2018) [27]	https://doi.org/10.1111/inm.12298 (accessed on 14 July 2022)	Australia	To investigate how empathy is developed and maintained when MHNs and service users disagree.	Community setting	Service users (mean age of 45 years) (n = 7)	Qualitative	Purposive sampling	One-on-one interviews		Thematic analysis	Showing empathy creates trust and reduces anger as well as paranoia. Empathy emerges when MHNs strive to understand the dispute from the service users’ viewpoints. Service users felt empathised with when MHNs listened, respected, and were non-judgmental.
Giménez-Díez D et al. (2020) [28]	https://doi.org/10.1111/jpm.12573 (accessed on 14 July 2022)	Spain	To examine the satisfaction of service users and their families with the nursing care provided through a hospital’s home care programme that promotes person-centred care	Mental health crisis assessment and treatment team	Service users (n = 20) Carers (n = 20)	A cross-sectional study including quantitative survey data and qualitative interview	Convenience sampling	One-on-one interviews	The CARE Q questionnaire measured the nurse’s behaviour and had a Cronbach’s alpha of 0.853. The CSQ-8 measured satisfaction with nursing care and had a Cronbach’s alpha of 0.85.	Descriptive statistics and framework analysis	Both service users and carers were satisfied with nursing care, but service users were more so. Service users’ satisfaction was associated with staff flexibility and experience. MHNs were regarded as professional, reliable, and caring.
Goodwin et al. (2006) [29]	https://doi.org/10.1111/j.1447-0349.2006.00413.x (accessed on 14 July 2022)	Australia	To examine how service users and carers participate in mental health care from the perspective of the carers	Mental health services (n = 2 centres)	Carers (n = 19)	Exploratory qualitative	Convenience sampling	Focus group		Content analysis	Service users and carers often have conflicting agendas. When service users and carers had competing interests, MHNs prioritised service users’ demands. Carers viewed conflicting agendas as barriers to care participation.
Goodwin et al. (2007) * [30]	https://doi.org/10.1080/01612840701354596 (accessed on 14 July 2022)	Australia	To explore carers’ attitudes and perceptions of their involvement in mental health care	A combination of bed- based and community units (n = 2 centres)	Service users (n = Not reported) Carers (n = not reported)	Exploratory qualitative	Convenience sampling	Focus group		Content analysis	Participants believed that mutual trust and respect were necessary for effective collaboration to occur. MHNs were compassionate and inclusive. Carers valued the accessibility of MHNs to both the service user and the carer.
Goodwin et al. (2007) * [31]	https://doi.org/10.1080/01612840701354612 (accessed on 14 July 2022)	Australia	To explore carers’ perceptions of their involvement in mental health care	A combination of bed- based and community units and team services (n = 2 centres)	Service users (n = Not reported)		Goodwin et al. (2007)	https://doi.org/10.1080/01612840701354612 (accessed on 14 July 2022)	Australia	To explore carers’ perceptions of their involvement in mental health care	A combination of bed- based and community units and team services (n = 2 centres)
Gray & Brown (2017) [32]	https://doi.org/10.1111/inm.12296 (accessed on 10 June 2022)	United Kingdom	To assess and contrast the service user and clinician perspectives about the practice of MHNs in promoting physical health in people with severe mental illness	Inpatient mental health, rehabilitation, and community settings (n = 2 centres)	Service users (mean age of 30 years) with schizophrenia bipolar disorder, Constipation and Hypertension (n = 15)	Qualitative	Convenience sampling	One-on-one interviews		Thematic analysis	Service users reported that MHNs rarely helped them with common adverse effects of medication. MHNs had no time to assist service users in changing their unhealthy habits. Service users expect MHNs to be more skilled in providing physical health care.
Gunasekara et al. (2014) [33]	https://doi.org/10.1111/inm.12027 (accessed on 10 June 2022)	Australia	To assess the perspectives of service users and carers on mental health nursing care.	Inpatient mental health setting	Service users (n = 10) Carers (n = 10)	Qualitative	Purposive sampling	One-on-one interviews		Thematic analysis	Service users expected MHNs to respect and educate them about their treatment. Carers wanted to be included in care planning and believed they could help the care team if appropriately supported. MHNs viewed advocating for service users as critical to building therapeutic relationships.
Happell et al. (2009) [34]	https://doi.org/10.1176/ps.2009.60.11.1527 (accessed on 14 July 2022)	Australia	To compare the level of satisfaction between service users receiving nurse-initiated care (experimental group) and those receiving treatment as usual (control group).	Mental health crisis assessment and treatment team	Service users with schizophrenia, schizoaffective disorder, depression, anxiety disorders (n = 103) Carers (n = not reported)	Quasi-experimental study	Service users were randomly assigned to receive care from the nurse practitioner candidate or to treatment as usual.	Survey	Content validity of the questionnaire was established in the study referenced by the authors. Cronbach’s alpha coefficient was 0.91.	Descriptive analysis	Participants’ perspectives were not adequately described. Service users and carers were satisfied with the mental health nursing care.
Happell & Palmer (2010) [35]	https://doi.org/10.3109/01612840.2010.488784 (accessed on 10 June 2022)	Australia	To assess the service users’ experiences and perspectives on the care they received from the Mental Health Nurses Incentive Program	Primary health care setting	Service users (n = 14)	A descriptive, exploratory qualitative study	Snowballing sampling	One-on-one interviews		Thematic analysis	Service users perceived MHNs as flexible, and this increased treatment compliance. Participants described nurses as having a broad knowledge of the health care system. Service users viewed the services offered by MHNs as providing more privacy than the public mental health system.
Horgan et al. (2021) [36]	https://doi.org/10.1111/inm.12768 (accessed on 10 June 2022)	Australia and Europe	To examine the views of mental health service users on the required qualities of a mental health nurse as an input for the development of a learning module	University and community settings (n = 7 centres)	Service users (n = 50)	Qualitative	Convenience sampling	Focus group		Thematic analysis	Service users valued MHNs who respected them, gave them hope and were non-judgemental. MHNs were expected to know the referral process and the organisations that assist service users. Service users desire MHNs to treat them with empathy and to assist them in coping with their challenges.
Jones et al. (2007) [37]	https://doi.org/10.1111/j.1365-2648.2007.04332.x (accessed on 14 July 2022)	United Kingdom	To assess the perspectives of service users towards the MHN prescribers	Mental healthcare organization	Service users (mean age of 43 years) (n = 12) ^a^	Qualitative	Purposive sampling	One-on-one interviews		Thematic analysis	The participants viewed supplemental prescribing by MHNs as satisfactory. They also felt that MHNs provided more in-depth descriptions of treatment alternatives. Service users perceived MHNs prescribers paid close attention to physical health.
Keogh et al. (2020) [38]	https://doi.org/10.1080/01612840.2020.1731889 (accessed on 10 June 2022)	Ireland	To find out how Irish travellers perceive the Traveller Mental Health Liaison Nurse (TMHLN)	Community mental health setting	Service users (n = 10)	Descriptive qualitative study	Convenience sampling	One-on-one interviews		Thematic analysis	Service users viewed the TMHLN as providing confidentiality and privacy, allowing them to feel at ease and discuss their issues. The TMHLN was described as kind, understanding, and trustworthy. Participants thought that they were assisted in resolving some of the social problems that were bothering them.
Kertchok (2014) [39]	https://doi.org/10.3109/01612840.2014.908439 (accessed on 10 June 2022)	Thailand	To examine the relationship between carers of people with schizophrenia and community MHNs	Community mental health setting	Carers (n = 17)	Grounded theory methodology	Purposive theoretical sampling	One-on-one interviews		Constant comparative methods	MHNs were described as kind. Carers felt MHNs provided them with information about care and were involved in the care of service users. MHNs encouraged carers to express their concerns and needs.
King et al. (2019) [40]	https://doi.org/10.3928/02793695-20190225-01 (accessed on 14 July 2022)	United States of America	To examine service users’ satisfaction with nursing care at an inpatient mental health unit.	Inpatient mental health setting	Service users aged between 18 to 91 years (n = 169)	Cross sectional	Convenience sampling	Survey	The Caring Behaviours Inventory-16 measured the competencies of nurses: Cronbach’s alpha coefficients = 0.968. The Client Satisfaction with Care measured service user satisfaction with care: Part 1, Cronbach’s alpha = 0.883; and Part 2, Cronbach’s alpha = 0.898	Descriptive statistics	Confidentiality of service user information, feeling safe in the ward, and receiving medication on time were all highly rated by service users. Service users were highly satisfied with the nursing care. However, the time spent with MHNs was less satisfactory for service users.
Koga et al. (2006) [41]	https://doi.org/10.1590/S0104-11692006000200003 (accessed on 14 July 2022)	Brazil	To assess mental health care in the Family Health Care Program through the views of service users and their carers	Community mental health setting	Service users aged between 21 to 70 years with anxiety, depression (n = 18) Carers aged between 20 to 74 years (n = 29)	Survey	Convenience sampling	Survey	Not reported.	Descriptive analysis	Service users reported that MHNs rarely explained medications. Carers perceived they did not receive enough information about medication. Service users reported that they received care when required.
Lees et al. (2014) [42]	https://doi.org/10.1111/inm.12061 (accessed on 14 July 2022)	Australia	To examine service users’ needs and experiences during suicidal crises, and the role of MHNs	Community and inpatient settings	Service users (average age of 41 years) with Suicidal crises (n = 9)	Qualitative	Convenience sampling	Interviews		Constant comparative, classical content analysis	Participants perceived MHNs as not interested in discovering what precipitated their suicidal crisis. They believed that MHNs abused their power. Service users felt they had few therapeutic interactions with MHNs.
Lessard-Deschênes & Goulet (2021) [43]	https://doi.org/10.1111/jpm.12800 (accessed on 14 July 2022)	Canada	To explore the therapeutic relationship in the context of involuntary treatment orders as perceived by service users	Inpatient mental health setting	Service users aged between 30 to 62 years with schizophrenia and bipolar disorder (n = 6)	Secondary data analysis of qualitative interviews	Convenience sampling	One-on-one interviews		Content analysis	The relationships between the service users and MHNs were superficial and never reached a therapeutic level. Service users viewed forced treatment as impeding the establishment of a trusting relationship. The safety-oriented strategy used by MHNs was considered abusive.
Lim et al. (2019) [44]	http://hdl.handle.net/20.500.11937/77779 (accessed on 10 June 2022)	Australia	Determine how service users perceive MHNs’ use of recovery-oriented care to address aggressive behaviour	Inpatient mental health setting	Service users aged 18 or older with mood, substance-related, post-traumatic stress disorders, schizophrenia & other psychotic disorders (n = 31)	Grounded theory methodology	Purposive and theoretical sampling	One-on-one interviews and Focus group		Constant comparative method	Participants stated that MHNs must treat them as individuals. Service users thought MHNs were quick to identify and judge them as potentially aggressive when they displayed negative emotions. Participants emphasised the importance of MHNs interacting positively with them.
McAllister et al. (2021) [45]	https://doi.org/10.1111/inm.12835 (accessed on 10 June 2022)	United Kingdom	To explore how engagement is experienced in acute units and the requirements of service users, carers, and clinicians to develop a collaborative intervention	Inpatient mental health setting	Service users aged between 18 to 64 years with psychotic, mood & personality disorders (n = 14) Carers aged between 18 to 64 years (n = 2)	Experience-based Co-design (EBCD) Qualitative study	Convenience sampling	One-on-one interviews, observations, and field notes		Thematic analysis	Service users and carers reported a lack of high-quality, person-centred, collaborative engagement. Service users and carers frequently felt that their concerns were not heard. All participants perceived that there was a need to strengthen nurse-patient engagement.
McCann & Clark (2008) [46]	https://doi.org/10.1111/j.1440-172X.2008.00674.x (accessed on 10 June 2022)	Australia	To explore the perceptions of service users with schizophrenia regarding unrestricted autonomous non-medical prescription of antipsychotic medications by MHNs	Community mental health settings (n = 3 centres)	Service users aged between 19 to 65 years with schizophrenia (n = 81)	Cross sectional	Non-probability sampling	Survey	The Factors Influencing Neuroleptic Medication Taking Scale measured service users’ views of prescription of antipsychotic medication by nurses. Cronbach’s alpha values were not reported	Descriptive analysis	More than half of the participants favoured MHNs having prescriptive authority. Participants under the age of 36 were more likely than those over the age of 36 to support allowing specially trained MHNs to stop prescribing medication. Many participants reported being satisfied with their relationship with MHNs.
McCann et al. (2012) [47]	https://doi.org/10.1111/j.1365-2702.2011.03836.x (accessed on 10 June 2022)	Australia	To assess the perspectives of first-time carers regarding how MHNs respond to them as carers of young people with first-episode psychosis.	Community mental health setting	Carers of service users with first-episode psychosis (n = 20)	Qualitative interpretative phenomenological analysis	Purposive sampling	Interviews		Interpretative phenomenological analysis	Two competing themes emerged. Carers perceived MHNs and other mental health professionals as accessible, attentive, and responsive to their needs. Second, carers believed some clinicians undervalued their role and excluded them from clinical decision-making about the young person.
McCloughen et al. (2011) [48]	https://doi.org/10.1111/j.1447-0349.2010.00708.x (accessed on 14 July 2022)	Australia	To assess if service users and nurses in a mental health rehabilitation setting had common understandings, attitudes, values, and experiences of collaboration	Mental health rehabilitation setting	Service users with severe mental illnesses (n = 18)	Mixed-method approach comprising focus groups and surveys	Purposive and convenience sampling	Focus group and survey	Not reported	Descriptive statistics and thematic analysis	Service users viewed collaboration as a partnership centred on the service users’ goals and wellness. They also perceived that active participation was essential. Collaboration requires effective communication and mutual recognition of one another’s expertise and skills.
Moll et al. (2018) [49]	https://doi.org/10.3928/02793695-20180305-04 (accessed on 14 July 2022)	Brazil	To examine the carers’ views and expectations of nursing care provided to mental health inpatients in a general hospital	Inpatient mental health setting	Carers (n = 10)	Descriptive–exploratory qualitative	Convenience sampling	Interviews		Content analysis	Carers were satisfied with the nursing care provided to their relatives. Establishing good interpersonal relationships with service users requires professionalism, care, and clinical competencies. Many carers had no higher expectations for nursing care than what was already being provided.
Montreuil et al. (2015) [50]	https://doi.org/10.3109/01612840.2015.1075235 (accessed on 10 June 2022)	Canada	To examine service users at risk of suicide and their carers’ perceptions of nursing care in paediatric mental health settings	Paediatric mental health inpatient, outpatient, and day hospital settings.	Service users aged between 11 to 14 years with suicide risk factors (n = 5) Carers (n = 5)	Exploratory qualitative design	Convenience sampling	One-on-one interviews, observation, debriefing sessions, and survey		Thematic analysis	Participants felt that helpful nursing care for service users and carers is based on nursing interventions that help them collaborate with MHNs. Service users and carers felt that MHNs could help carers by being accessible and reassuring. All participants thought MHNs were vital in controlling shared and private places.
Pitkänen et al. (2008) [51]	https://doi.org/10.1016/j.ijnurstu.2008.03.003 (accessed on 10 June 2022)	Finland	To assess how the nursing care in acute mental health inpatient units impacts the quality of life of service users.	Inpatient mental health settings (n = 7 centres)	Service users with schizophrenia, schizotypal disorder (n = 35)	Explorative descriptive qualitative study	Convenience sampling	One-on-one interviews		Content analysis	Service users perceived that MHNs empowered them by allowing them to make choices. Participants valued the opportunities provided by MHNs to participate in recreational activities. Service users expected MHNs to provide them with information about their illnesses and treatment.
Rask & Brunt (2006) [52]	https://doi.org/10.1111/j.1447-0349.2006.00409.x (accessed on 14 July 2022)	Sweden	To determine the perspective of service users on the frequency and significance of verbal and social nursing interactions in inpatient mental health settings.	Inpatient mental health setting	Service users aged between 20 to 46 years with Mood disorders, sexual disorders, substance abuse, personality disorders, schizophrenia & other psychotic disorders (n = 20)	Cross sectional	Convenience sampling	Survey	Verbal and Social Interactions (VSI) measured the frequency and importance of nursing interactions. Cronbach’s alpha = 0.95).	Descriptive statistics	Service users thought it was important that MHNs explain what they could do to help themselves. Participants felt MHNs frequently encouraged them to learn new things. Service users thought it was vital for them to discuss their feelings with MHNs.
Romeu-Labayen et al. (2022) [53] ^#^	https://doi.org/10.1111/jpm.12766 (accessed on 14 July 2022)	Spain	To explore how service users with borderline personality disorder perceive the role of MHNs in building a positive therapeutic relationship	Community mental health setting	Service users with borderline personality disorder (n = 12)	Qualitative descriptive design	Purposive sampling	One-on-one interviews		Thematic analysis	Service users trusted the MHNs more after being listened to and seeing empathy. Inspiring service users to change and heal was positively received by the service users. Participants saw MHNs as genuine when they used humour and felt accepted.
Rose et al. (2015) [54]	https://doi.org/10.1017/S2045796013000693 (accessed on 14 July 2022)	United Kingdom	To examine the views and experiences of service users about life in an acute mental health ward	Inpatient mental health setting	Service users with schizophrenia and psychotic disorders (n = 37)	In-depth secondary analysis of focus group data	Purposive sampling	Focus group		Thematic analysis	The consensus among service users was that the ward was not therapeutic. Service users perceived MHNs as unavailable and uncaring. Participants felt forceful interventions were unneeded and overbearing.
Rydon (2005) [55]	https://doi.org/10.1111/j.1440-0979.2005.00363.x (accessed on 14 July 2022)	New Zealand	To identify the attitudes, knowledge, and skills expected of mental health nurses	Support groups for users of mental health services	Service users (n = 21) Carers (n = not reported)	Qualitative descriptive methodology	Convenience sampling	Focus group		Thematic analysis	Service users highly regarded the therapeutic work of MHNs. Participants did not always experience therapeutic interactions in their encounters with MHNs. Service users identified positive attitudes towards service users as an important characteristic of MHNs.
Santangelo et al. (2018) [56]	https://doi.org/10.1111/inm.12317 (accessed on 14 July 2022)	Australia	To create a theoretical model of mental health nursing practice that focuses on identifying qualities that contribute to favourable outcomes for service users	Community	Service users (n = 5)	Constructivist grounded theory	Purposive sampling	One-on-one interviews		Constant comparative method	Positive therapeutic relationships were considered essential in mental health nursing by service users. MHNs were described as spending more time with service users than any other professionals. Participants perceived that MHNs provided holistic care.
Saur et al. (2007) [57]	https://doi.org/10.1177/1078390307301938 (accessed on 14 July 2022)	United States of America	To find out how satisfied service users are with the treatment they received from MHN specialists and their preferences for future depression therapy	Primary health care setting	Service users aged 60 years or older with depression (n = 105)	Cross sectional	Random sampling	Survey	Questionnaire measured service user satisfaction with nursing care. Psychometric properties not reported.	Descriptive statistics	The majority of service users expressed satisfaction with the quality of care provided by MHNs. Service users viewed the therapeutic relationship with MHNs as highly positive. Most of the service users are willing to seek care from MHNs in the future.
Schneidtinger et al. (2019) [58]	https://doi.org/10.1111/jcap.12245 (accessed on 14 July 2022)	Austria	To investigate how adolescents receiving mental health services experienced personal recovery	Community mental health setting	Service users aged between 15 to 19 years (n = 9)	Qualitative exploratory study	Self-selection sampling	One-on-one interviews		Content analysis	Service users reported that MHNs aided their recovery by teaching them coping strategies. The participants perceived that MHNs were always available for conversations. The presence of nurses gave service users a sense of security.
Shattell et al. (2007) [59]	https://doi.org/10.1111/j.1447-0349.2007.00477.x (accessed on 14 July 2022)	United States of America	To explore service users’ experience of the therapeutic relationship	Community Mental Health setting	Service users aged between 21 to 65 years with mood, anxiety, antisocial personality disorders and schizophrenia (n = 20)	Secondary analysis of qualitative	Purposeful sampling	One-on-one interviews		Existential phenomenological approach	Service users had positive therapeutic relationships with MHNs. Participants thought building therapeutic relationships required providing psychological support, information, and referral recommendations to service users. Service users expect MHNs to know them as people, not as diagnoses or statistics.
Sinclair et al. (2006) [60]	https://doi.org/10.1136/emj.2005.033175 (accessed on 14 July 2022)	United Kingdom	To examine the perception of service users about the care provided by MHNs in an accident and emergency department	Mental health crisis assessment and treatment team (n = 2 centres)	service users with Mental health disorders (n = 511)	Crossover design	Convenience sampling	Survey	Survey measured provision of information, care and treatment received. Psychometric properties not reported.	Descriptive statistics, linear and multinomial regression, and content analysis	Levels of satisfaction recorded were high for all service users with no significant differences between intervention and non-intervention periods. There were no concerns about MHNs.
Stenhouse (2011) [61]	https://doi.org/10.1111/j.1365-2850.2010.01645.x (accessed on 14 July 2022)	United Kingdom	To obtain insight into the experience of being a service user on an acute inpatient mental health ward	Inpatient mental health setting	Service users aged between 18 to 65 years (n = 13)	Qualitative	Convenience sampling	One-on-one interviews		Holistic analyses	Service users anticipated that MHNs would interact with them. However, MHNs did not approach them to initiate conversation, which was viewed as disinterest and a lack of caring. Participants believed nurses were frequently too busy to engage in conversation.
Stewart et al. (2015) [62]	https://doi.org/10.1111/inm.12107 (accessed on 14 July 2022)	United Kingdom	To explore service users’ views of the personal and professional qualities of MHNs and how these contribute to the ward environment	Inpatient mental health setting (n = 16 centres)	Service users (62% were under 40 years) (n = 119)	Qualitative	Random sampling	One-on-one interviews		Thematic analysis	Service users recognised that MHNs have challenging and demanding work. Service users frequently expressed anger and hopelessness about their ward experience. Participants were frustrated that MHNs could not comprehend or sympathise with their concerns.
Terry (2020) [63]	https://doi.org/10.1111/inm.12676 (accessed on 14 July 2022)	United Kingdom	To assess how mental health nursing was perceived by services users.	Community setting	Service users (n = 13)	Qualitative	Purposive sampling	One-on-one interviews and focus group		Thematic analysis	Participants labelled MHNs as “bridging the gap’ because they require various skills to meet the needs of different people. Service users viewed listening and helping as the most important nursing roles. Participants valued their therapeutic relationships with MHNs.
Testerink et al. (2019) [64]	https://doi.org/10.1111/ppc.12275 (accessed on 14 July 2022)	Netherlands	To explore the experiences of carers with nursing care provided to their relatives during admission to closed wards for mania	Inpatient mental health setting (n = 3 centres)	Carers of service users with mania (n = 9)	*Descriptive phenomenological* design	Convenience sampling	One-on-one interviews		Stevick-Colaizzi-Keen method	Some carers described nurses as polite and helpful, while others perceived them as lacking passion for their work. Participants were disappointed that nurses were not listening to their advice. Carers desired to be involved in care planning, and some even requested to participate.
Wand & Schaecken (2006) [65]	https://doi.org/10.5172/conu.2006.21.1.14 (accessed on 14 July 2022)	Australia	To assess the role of a mental health liaison nurse in the emergency department in Australia	Mental health crisis assessment and treatment team	Service users aged between 19 to 82 years (n = 59)	Cross sectional	Convenience sampling	Survey	Consumer satisfaction survey designed by Gillette et al. (1996) measured service users’ satisfaction levels with nursing care. We could not retrieve the referenced study (Gillette et al., 1996) to check the validity of the questionnaire	Descriptive statistics and thematic analysis	Most respondents acknowledged that the MHN was too willing to listen and provide emotional support. Service users reported that the MHN had expert knowledge of mental illnesses. Most of the service users described the MHN as empathetic, compassionate, and friendly.
Wilson (2010) [66]	https://doi.org/10.1111/j.1365-2850.2010.01586.x (accessed on 14 July 2022)	United States of America	To examine the views of service users about the competencies of MHN in providing culturally congruent care	Primary health care setting	Service users aged between 18 to 65 years (n = 15)	Qualitative descriptive study	Snowballing sampling	One-on-one interviews and qualitative survey		Thematic content analysis	Service users perceived that medication administration was the most important nursing intervention. Participants were allowed to participant in their religious practice. Service users struggled to identify culture-specific nursing interventions that improved their mental health.
Wortans et al. (2006) [67]	https://doi.org/10.1111/j.1365-2850.2006.00916.x (accessed on 14 July 2022)	Australia	To establish the feasibility of implementing a nurse practitioner role in a variety of settings in Victoria, Australia	Mental health crisis assessment and treatment team (n = 4 centres)	Service users with schizophrenia, personality disorder and situational crises (n = 7)	Qualitative exploratory study	Convenience sampling	One-on-one interviews		Thematic content analysis	Except for one person, every participant expressed unequivocal support for the role of the nurse practitioner candidate. All participants indicated that they could relate to the nurse practitioner more easily than doctors. Service users reported that they would not hesitate to seek care and treatment from a nurse practitioner in the future.

Notes: * Study with results reported across two articles, ^#^ Studies first published in 2021, ^a^ In the abstract, 12 service users participated in the study, but in the method section, 11 service users agreed to participate. Discussed the discrepancy between the abstract and the methods with the corresponding author, and he said he could not remember. The corresponding author is one of the authors in this review. The number of centres was only reported where there are multiple centres.

**Table 2 ijerph-19-11001-t002:** Summary of the characteristics of the included studies.

Study Characteristics	Number of Studies [%]
Study design	
Qualitative studies	38 [78%]
Cross sectional	7 [14%]
Quasi-experimental	1 [2%]
Mixed methods	2 [4%]
Consultation	1 [2%]
Settings	
Inpatient	12 [24%]
Community	31 [63%]
Mixed services	6 [12%]
Population	
Adults	21 [43%]
Children and adolescents	2 [4%]
Elderly	1 [2%]
Not reported	25 [51%]
Study centres	
Multiple centres	13 [27%]
Single centres	36 [73%]
Countries where studies were conducted	
Australia	16 [33%]
United Kingdom	11 [22%]
United States of America	4 [8%]
Canada	3 [6%]
Ireland	3 [6%]
Norway	2 [4%]
Spain	2 [4%]
Brazil	2 [4%]
Thailand	1 [2%]
Finland	1 [2%]
Sweden	1 [2%]
New Zealand	1 [2%]
Netherlands	1 [2%]

**Table 3 ijerph-19-11001-t003:** Quality assessment for cross-sectional and experimental studies using the effective public health practice project quality assessment tool.

Study Author	Digital Object Identifier (DOI)	Selection Bias	Study Design	Confounders	Blinding	Data Collection Method	Withdrawals and Dropouts	Global Rating
King et al. (2019) [40]	https://doi.org/10.3928/02793695-20190225-01 (accessed on 20 June 2022)	Weak	Weak	Weak	Weak	Strong	Moderate	Weak
Koga et al. (2006) [41]	https://doi.org/10.1590/S0104-11692006000200003 (accessed on 20 June 2022)	Weak	Weak	Weak	Weak	Weak	Moderate	Weak
Happel et al. (2009) [34]	https://doi.org/10.1176/ps.2009.60.11.1527 (accessed on 20 June 2022)	Moderate	Moderate	Weak	Weak	Strong	Moderate	Weak
McCann & Clark (2008) [46]	https://doi.org/10.1111/j.1440-172X.2008.00674.x (accessed on 20 June 2022)	Weak	Weak	Weak	Weak	Moderate	Moderate	Weak
Rask & Brunt (2006) [52]	https://doi.org/10.1111/j.1447-0349.2006.00409.x (accessed on 20 June 2022)	Weak	Weak	Weak	Weak	Strong	Moderate	Weak
Saur et al. (2007) [57]	https://doi.org/10.1177/1078390307301938 (accessed on 20 June 2022)	Moderate	Weak	Weak	Weak	Weak	Moderate	Weak
Sinclair et al. (2006) [60]	https://doi.org/10.1136/emj.2005.033175 (accessed on 20 June 2022)	Weak	Weak	Weak	Weak	Weak	Moderate	Weak
Wand et al. (2006) [65]	https://doi.org/10.5172/conu.2006.21.1.14 (accessed on 20 June 2022)	Weak	Weak	Weak	Weak	Weak	Moderate	Weak

**Table 4 ijerph-19-11001-t004:** Risk of bias assessment for mixed method studies using the mixed methods appraisal tool [MMAT].

Study Authors	Digital Object Identifier (DOI)	Criteria 1	Criteria 2	Criteria 3	Criteria 4	Criteria 5
Giménez-Díez D et al. (2020) [28]	https://doi.org/10.1111/jpm.12573 (accessed on 20 June 2022)	No	Yes	Yes	Can’t tell	Yes ^a^
McCloughen et al. (2011) [48]	https://doi.org/10.1111/j.1447-0349.2010.00708.x (accessed on 20 June 2022)	Yes	Yes	Yes	Can’t tell	No

Notes: ^a^ We contacted the corresponding author regarding the response rate. The corresponding author reported that the response rate was 100%.

## Data Availability

Data is contained within the article and Appendix A.

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
