# Peer review of "Service User and Carer Views and Expectations of Mental Health Nurses: A Systematic Review"

_ijerph, 2022, doi:10.3390/ijerph191711001_

Round 1

Reviewer 1 Report

Thank you for the opportunity to review this update of a systematic review. It is important to update and expand this review about service users' and cares view about mental health nurses' work. In general, I found the methodological quality of the paper to be very good. The discussion section is something the authors could have a second look at to make it more complete. Please find my comments below.

Title: maybe consider adding carers to the title? As they are an integral part of this review, I suggest including them more strongly in different parts of the manuscript.

Material and methods

2.9 Data synthesis: Could you elaborate on this a little bit more? As most of the included studies were qualitative, it would be good to understand how you summarised your data with numbers (key findings or something else?). The approach for narrative synthesis could be added here as well. Was it inductive? It seems that one study can be classified under different themes (19, for example) - this could be explained here.

Discussion

- This starts with 'Authors should' - please remove

- There is room to expand this section and make it more in-depth. For example, views of cares should be discussed. Are they similar to service users'? It would be interesting to discuss also that even though service users a in general satisfied with the care, they (and carers) have negative views about the work of mental health nurses. Are they important differences in those studies, for example? Why views and expectations have not changed in the past 15 years? What could be the future research directions in this area and message to policymakers?

Reviewer 2 Report

Dear Authors,

Thank you for the possibility of reading your manuscript.

Here are some comments:

-The Document 4S: Summary of included studies, include relevant informationt and it should be in the main document

-Include the Table 1. Search strategy: Ovid MEDLINE(R) as supplementary material.

-Include in tables 3, 4 and 5 the reference number of each study.

-Revise the order of the material and methods subsections.

Reviewer 3 Report

1. Introduction

In the introduction you have to justify much better the importance of doing a review about the views and expectations of mental health nurses. In the introduction you have only shown the weaknesses of a previous review. The authors need to show the reader the importance of doing a review of those concepts in that workforce.

The last paragraph (line 68 to 72) should not be there but in the method section.

2.Method

It is not important that they have recorded their research there. They are confusing. They should do it in Prospero (https://www.crd.york.ac.uk/prospero/) or similar.     

3. Results

It is very difficult to interpret the results as they are presented. Reviews usually present a summary of the studies reviewed in one or more summary tables. The reviewer would suggest that, to make it easier for the reader to understand their work, they should put the table in "supplementary document 4" in the text.

The tables with the quality of the reviewed papers, on the other hand, do not provide relevant information and should therefore be included in supplementary documentation.

3.3. It is not clear to the reviewer whether the work presented is a systematic or narrative review.

3.4. It would have been very interesting, visual and highly recommended, if this information had been put in a table.

The information presented needs to be better organized. They are mixing variables related to quality of care (satisfaction, user-centered care) with personal variables or with external elements such as prescribing capacity.

4. Discussion

Further discussion of their results is needed. 
